# PHEV infection: A promising model of betacoronavirus-associated neurological and olfactory dysfunction

**Junchao Shi[1]☯, Zi Li[1]☯, Jing Zhang[1], Rongyi Xu[1], Yungang Lan[1], Jiyu Guan[1], Rui Gao[1], Zhenzhen Wang[1], Huijun Lu[1], Baofeng Xu[2], Kui Zhao[1], Feng Gao[1], Wenqi He[1] \***

**1** State Key Laboratory for Zoonotic Diseases, Key Laboratory for Zoonosis Research of the Ministry of Education, College of Veterinary Medicine, Jilin University, Changchun, China, **2** Department of Neurosurgery, the First Hospital of Jilin University, Changchun, China

☯ These authors contributed equally to this work.
\* hewq@jlu.edu.cn

**Data Availability Statement:** All relevant data are within the manuscript and its Supporting Information files.

## Abstract

Porcine hemagglutinating encephalomyelitis virus (PHEV) is a highly neurotropic coronavirus belonging to the genus *Betacoronavirus*. Similar to pathogenic coronaviruses to which humans are susceptible, such as SARS-CoV-2, PHEV is transmitted primarily through respiratory droplets and close contact, entering the central nervous system (CNS) from the peripheral nerves at the site of initial infection. However, the neuroinvasion route of PHEV are poorly understood. Here, we found that BALB/c mice are susceptible to intranasal PHEV infection and showed distinct neurological manifestations. The behavioral study and histopathological examination revealed that PHEV attacks neurons in the CNS and causes significant smell and taste dysfunction in mice. By tracking neuroinvasion, we identified that PHEV invades the CNS via the olfactory nerve and trigeminal nerve located in the nasal cavity, and olfactory sensory neurons (OSNs) were susceptible to viral infection. Immunofluorescence staining and ultrastructural observations revealed that viral materials traveling along axons, suggesting axonal transport may engage in rapid viral transmission in the CNS. Moreover, viral replication in the olfactory system and CNS is associated with inflammatory and immune responses, tissue disorganization and dysfunction. Overall, we proposed that PHEV may serve as a potential prototype for elucidating the pathogenesis of coronavirus-associated neurological complications and olfactory and taste disorders.

## Author summary

PHEV, a neurotropic porcine betacoronavirus (β-CoV), primarily infects and replicates in the respiratory tract and CNS in suckling pigs. Neurological complications and anosmia (i.e., inability to perceive odor or loss of olfactory function) are common clinical features in coronavirus-induced diseases, however the underlying mechanisms remain unclear. In this study, we investigated the pathogenesis of neurological and olfactory dysfunction in a murine model of PHEV infection. The data underscore that PHEV invades the CNS via

**Funding:** This work was supported by the National Natural Science Foundation of China (grant numbers 32172805, 32172828, 31902262, and 31872446); the Scientific and Technological Project of Jilin Province (grant numbers 20210202041NC to WH), the Youth Science and Technology Talent Support Project of Jilin Province (grant numbers QT202015 to ZL). The funder had no role in the study design, data collection and interpretation, or the decision to submit the work for publication.

**Competing interests:** The authors have declared that no competing interests exist.

the olfactory and trigeminal nerves, and that anosmia and neurological manifestations are associated with direct OSNs infection and neuroimmune inflammation. The utilization of PHEV prototype will provide a platform for future studies on the neuroinvasion and neuropathogenesis of human pathogenic coronaviruses.

## Introduction

Coronavirus disease 2019 (COVID-19) is caused by the newly emerged *betacoronavirus* (*β*-CoV) severe acute respiratory syndrome coronavirus 2 (SARS-CoV-2) and has had global impacts on public healthcare systems and economies [1,2]. In addition to airway and pulmonary symptoms, reduction or sudden loss of smell or taste has been reported in approximately half of all COVID-19 patients [3–5]. Furthermore, a wide range of central and peripheral neurological symptoms have been observed in patients with severe disease [6,7], suggesting that SARS-CoV-2 may target cells within the central nervous system (CNS) [8]. Currently, few animal models of COVID-19-associated anosmia, ageusia, and SARS-CoV-2 neuroinvasion are available [9–11]. The use of other *β*-CoVs, such as murine hepatitis virus (MHV), has been proposed as an approach for simulating several of the major characteristics of human pathogenic coronaviruses infection, including SARS-CoV-2-induced acute lung injury and systemic symptoms [12,13].

Porcine hemagglutinating encephalomyelitis virus (PHEV), along with SARS-CoV-2 and MHV, is a member of the genus *Betacoronavirus* within the family *Coronaviridae* and order *Nidovirales* [14]. After replicating in the upper respiratory tract, some PHEV strains also cause influenza-like symptoms (ILS) in adult pigs [15]. Notably, PHEV exhibits typical neurotropism and is currently the only known neurotropic coronavirus capable of infecting pigs [14]. Naturally, PHEV infects nasal epithelial cells and the tonsils in the respiratory tract, and then the virus propagates from the peripheral nerves to the CNS [14]. Clinical signs include encephalomyelitis, vomiting and wasting disease (VWD), and ILS. Encephalomyelitis in suckling pigs caused by PHEV infection was first reported in Canada in 1957, and the causative agent was first isolated in 1962 [16,17]. In 1969, another clinical type of PHEV-induced VWD in suckling piglets was observed in England [18]. Both clinical forms were experimentally reproduced in neonatal pigs using PHEV isolates from the same farm [19]. Generally, clinical manifestations of encephalomyelitis and VWD are age-dependent and reported frequently in piglets under 4 weeks old, with mortality rates reaching 100% [14,15,20]. However, an acute outbreak of ILS-like respiratory disease in adult exhibition swine was reported in the USA in 2015, and PHEV was identified as the causative agent [15]. Although only a few reports of PHEV outbreaks have been documented, they are devastating due to the lack of vaccines and effective countermeasures [14]. Furthermore, subclinical circulation of PHEV has been reported in many countries according to serological investigation, further emphasizing the significance of PHEV in pig farming worldwide [14,21].

In recent years, researchers have studied the pathogenesis of PHEV from multiple perspectives using mouse, rat and in vitro nerve cell models. Neural cell adhesion molecule (NCAM) interacts with PHEV, promoting entry into nerve cells [22,23]. In addition, cell-surface glycans, i.e., sialic acid (SA) and heparan sulfate (HS), act as attachment factors for PHEV in nerve cells [24]. Clathrin-mediated endocytosis (CME) and the endosomal system of neurons are hijacked by PHEV for virus intracellular trafficking [25]. Meanwhile, PHEV activates integrin α5β1-FAK-Cofilin signaling to induce rearrangement of the cytoskeleton, which in turn provides energy for the intracellular transport of virions [26]. In PHEV-infected neurons,

progeny virions bud and assemble in smooth-surfaced vesicles originating from endoplasmic reticulum–Golgi intermediate compartments and are then released from the cells by the biosynthetic secretory pathway [27]. The vesicle-mediated secretory pathway mediates the transsynaptic transmission of PHEV between neurons.

In the CNS, PHEV is mainly located in the neuronal soma and processes in the cerebral cortex, brain stem and spinal cord [28]. Neurodegenerative changes, such as axonal dysplasia, unstable dendritic spine formation, and irregular swelling and disconnection in neurites, are linked to the Ulk1-TrkA-NGF-Rab5 signaling pathway [29,30]. Furthermore, PHEV-induced neurodegeneration is related to lysosome dysfunction and endoplasmic reticulum (ER) stress [31–33], similar to the pathogenesis of human neurodegenerative diseases such as Parkinson's disease, frontotemporal degeneration and neuronal lipofuscinosis, indicating that PHEV might be a useful model virus to study the mechanisms of human neurodegenerative disease [33].

In this paper, we investigated the neuroinvasiveness of PHEV in BALB/c mice and proposed the potential application of intranasal PHEV infection in BALB/c mice as a model for investigating the pathogenesis of $\beta$-CoV-induced anosmia, ageusia, and neurological complications. PHEV-infected mice exhibit significant olfactory and gustatory dysfunction, with effective virus replication, robust inflammation, and functional impairment of the nasal epithelium and CNS. A better understanding of the neuroinvasion route and underlying mechanisms of the neuropathogenesis of PHEV after transport from the upper respiratory tract to and within the CNS will provide important insights into the development of antiviral countermeasures tailored to this specific host compartment for other neurotropic coronaviruses.

## Results

### Clinical manifestations in PHEV-inoculated BALB/c mice

Six-week-old (6w) and three-week-old (3w) BALB/c mice of both sexes were intranasally challenged with 20 μL of PBS or a 50% tissue culture infective dose (TCID$_{50}$) of $10^{3.96}$ PHEV (strain CC14) for behavioral experiments and neuroinvasion experiments. These mice were monitored daily for survival, weight change, and clinical symptoms (Fig 1A). Viral inoculation resulted in 100% mortality at day 5 post-infection (dpi) in PHEV-infected 3w mice and at 6 dpi in PHEV-infected 6w mice (Fig 1B). PHEV-infected mice exhibited a substantial decrease in body weight at 3 dpi in both age groups (Fig 1C) and began to display signs of sickness at 3 dpi in PHEV-infected 3w mice and 4 dpi in PHEV-infected 6w mice, including lethargy, delayed movement, and vocalizations (Fig 1D). Neurological signs such as ruffled fur, hunchback posture, tremors, and ataxic gait were detected at 4 dpi in PHEV-infected 3w mice and 5 dpi in PHEV-infected 6w mice (Fig 1D). No differences in death, weight loss, or clinical symptoms caused by PHEV were observed between sexes but between ages (S1A–S1C Fig). Collectively, these results suggest that BALB/c mice are susceptible to PHEV infection and that infected mice show obvious clinical manifestations.

### PHEV infection induces anosmia and ageusia in mice

Given that 6w mice have fully developed organs and have reached sexual maturity, we selected 6w mice for behavioral studies, although PHEV efficiently infects 3w and 6w mice. We performed a sucrose preference test at 1–3 dpi in 6w mice to assess taste function (Fig 2A–2C). No significant difference in the total intake of water overnight was observed between mock and PHEV-infected mice of both sexes (Fig 2B). As expected, mock-infected mice preferred 1% sucrose water to regular water, while infected mice of either sex showed no preference for sucrose-complemented water (Fig 2C). It indicated that PHEV-associated ageusia occurred. Smell function was evaluated by performing a series of behavioral experiments, as previously

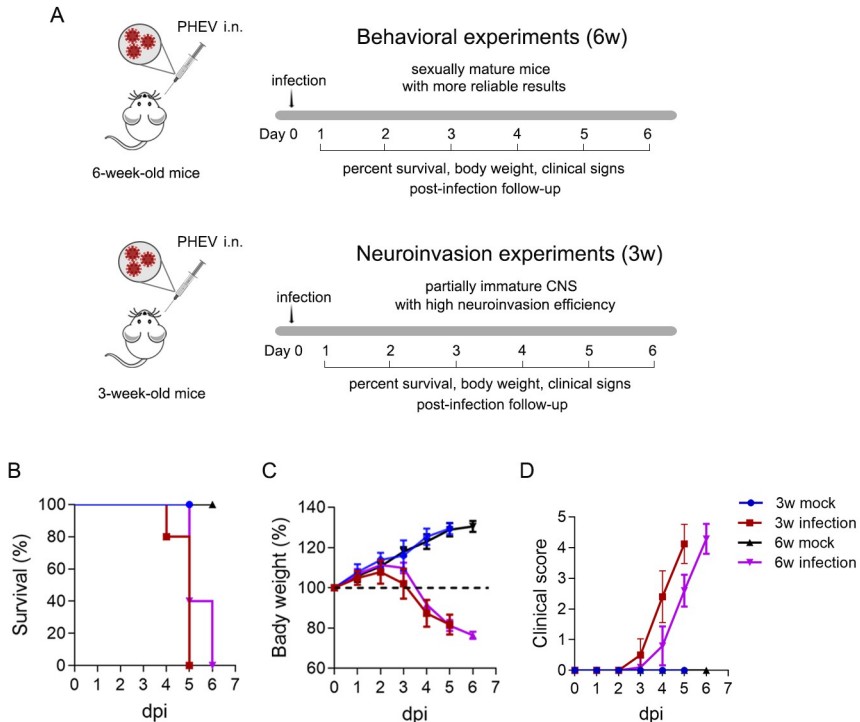

**Fig 1. Intranasal inoculation of PHEV in BALB/c mice results in lethal infection.** Three-week-old (3w) and six-week-old (6w) BALB/c mice were mock-infected (n = 3/sex/age) or intranasally inoculated (n = 5/sex/age) with $10^{3.96}$ TCID$_{50}$ PHEV. (A) Schematic diagram of the study design and workflow. Mice were monitored daily for survival (**B**), relative weight change (**C**), and clinical signs (**D**). Statistical analyses were performed using log-rank (Mantel–Cox) tests (B), Wilcoxon matched-pairs rank test (C), and one-way ANOVA, two-tailed Student's t test (D). Data are representative of three replicate experiments and are shown as the means ± SD.

reported [34]. First, the buried food finding test indicated that infected mice of both sexes took longer time to find the hidden food (Fig 2D and 2E), and a significant proportion of the mice (30% of males vs. 43% of females at 3 dpi) were unable to locate the food at the end of the test (Fig 2F). All infected mice were able to find visible food (S2 Fig), indicating that the delay in locating the buried food was not due to sickness-related behavior, vision impairment, or loco-motor deficiency. Second, male mice were exposed to 2-ml Eppendorf tubes filled with either female or male dander (Fig 2G). Male mice were attracted to female dander when olfaction was normal, but we found that the female dander was less attractive to PHEV-infected male mice (Fig 2H). Third, female mice were provided bedding from their home cage ('familiar scent') and a different foreign cage ('novel scent') (Fig 2J). Mice with normal olfaction pre-ferred the novel bedding, while PHEV-infected female mice spent less time near the foreign bedding (Fig 2K). Preference indices further confirmed anosmia in PHEV-infected mice of both sexes (Fig 2I and 2L). These results suggest that PHEV infection induces obvious smell and taste impairments in mice.

## Tropism of PHEV to neurons in the mouse CNS

The 3w and 6w BALB/c mice were intranasally inoculated with PHEV (strain CC14), and viral genomic RNA and infectious virus particles were detected at 5 dpi to determine the target

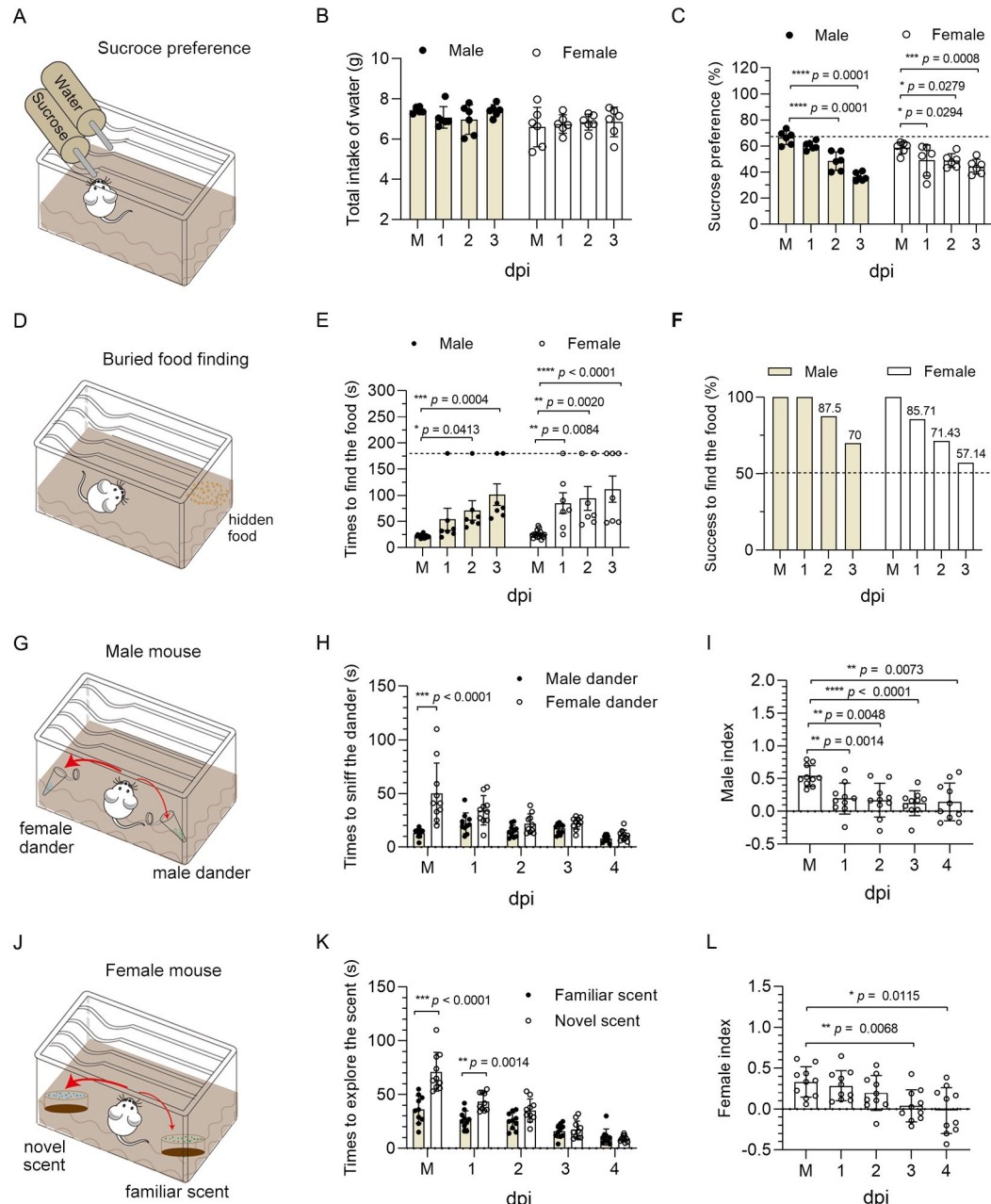

**Fig 2. PHEV-associated anosmia and ageusia in BALB/c mice.** The 6w male and female BALB/c mice were intranasally inoculated with $10^{3.96}$ TCID$_{50}$ PHEV or mock-infected with PBS (M), and a series of behavioral experiments was performed to assess the olfactory and taste functions in mock and PHEV-infected mice. **(A)** Schematic diagram of the sucrose preference test (n = 6 per sex). **(B)** Total intake of water overnight. **(C)** Sucrose preference of mock and PHEV-infected mice at 1–3 dpi. Each circle represents a mouse. **(D)** Schematic diagram of the buried food finding test (n = 7 per sex). **(E)** The time it took for male and female mice to find buried food. The dashed line represents the time limit of 3 min. **(F)** Percentage of mice that successfully found buried food within 3 min. **(G)** Schematic diagram of the social scent-discrimination test for male mice (n = 10). **(H)** Time that male mice spent sniffing male or female dander. (**I**) Preference indices for male mice. **(J)** Schematic diagram of the social scent-discrimination test for female mice (n = 10). **(K)** Time that female mice spent exploring familiar or novel scents. **(L)** Preference indices for female mice. *P* values were calculated by two-way ANOVA (**B, C, E, H** and **K**) and the two-tailed Mann–Whitney U test (**I** and **L**). Data are representative of three replicate experiments and are shown as the means ± SD.

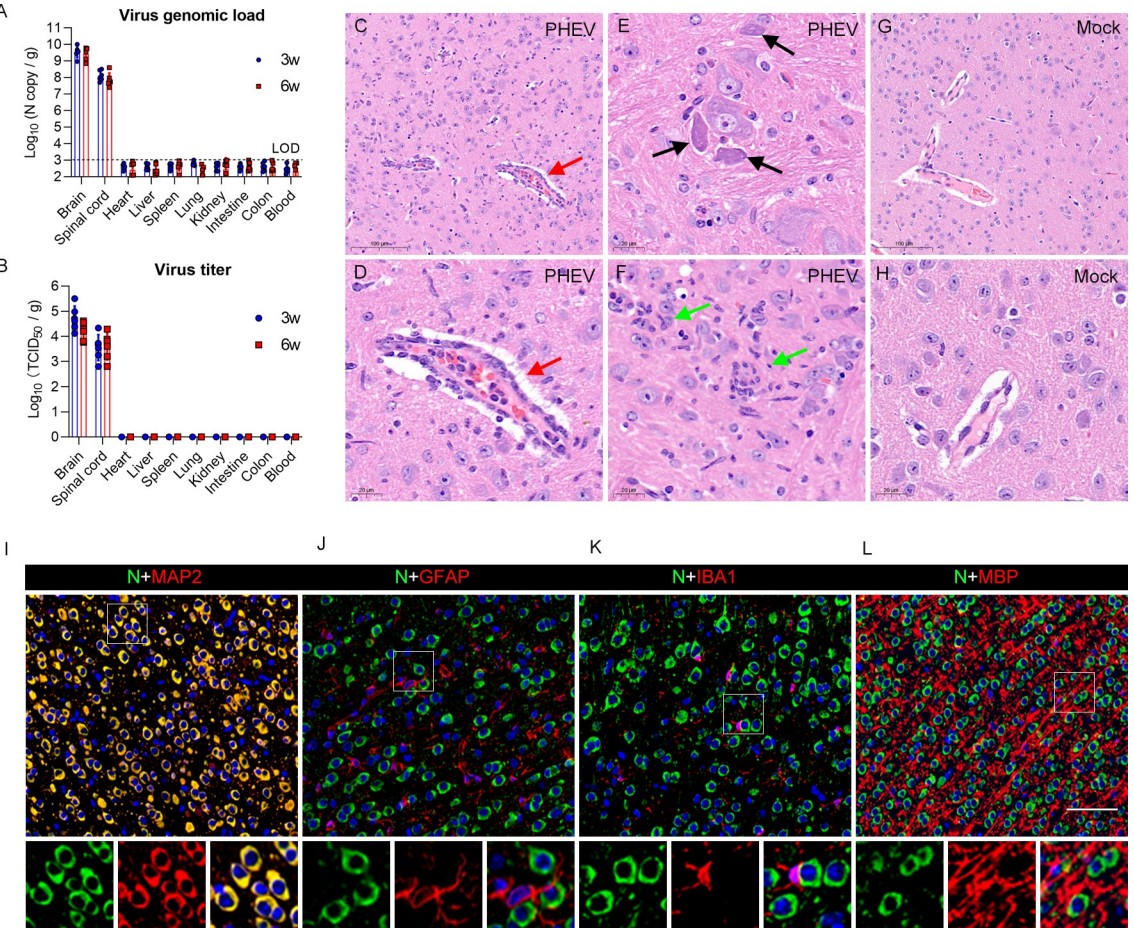

**Fig 3. Brain damage and viral tropism in the CNS of PHEV-infected mice. (A-B)** The 3w and 6w BALB/c mice were inoculated with $10^{3.96}$ TCID$_{50}$ PHEV and samples were collected at 5 dpi for qRT–PCR and viral titer determination (n = 6). **(A)** Viral genome loads were monitored in different organs and blood. The limit of detection (LOD) is shown with a dashed line. **(B)** Infectious viral titers were detected in different organs (n = 6). **(C-L)** The 3w BALB/c mice were intranasally inoculated with $10^{3.96}$ TCID$_{50}$ PHEV. Mice were euthanized at 5 dpi, and brain samples were harvested for histopathological examination. **(C-H)** H&E staining of brain sections from control and PHEV-infected mice. **(C-D)** Lymphocytic perivascular cuffing (red arrows). **(E)** Dying neurons undergoing degeneration (black arrows). **(F)** Microglial nodules (green arrows). **(G-H)** Brains from mock-infected mice. **(I-L)** Immunofluorescence images of PHEV-infected neurons in 3w BALB/c mice. MAP-2, GFAP, IBA1, and MBP are markers for neurons, astrocytes, microglia, and oligodendroglia, respectively. Scale bars, 100 μm **(C, G)**, 20 μm **(D-F, H)**, and 50 μm **(I-L)**.

organs for PHEV infection. PHEV preferentially infects the brain and spinal cord, while no virus antigens or infectious virions were detected in peripheral organs, including heart, liver, spleen, lung, kidney, small intestine, colon, and blood (Fig 3A and 3B). Consistent with these findings, no significant pathological changes in the liver, spleen, kidney, small intestine, or colon were observed in either mock or PHEV-infected mice (S3 Fig).

Because of the higher efficiency of PHEV neuroinvasion in younger mice, we used immature 3w mice to determine the target cells for PHEV infection in the CNS. When neurological symptoms were visible, mice were euthanized and brains were harvested at 5 dpi. Histopathologically, perivascular cuffing was observed in the cerebral parenchyma of PHEV-infected mice due to lymphocyte infiltration (Fig 3C and 3D, red arrow). Degenerated neurons had basophilic, angular, and shrunken cell bodies with contracted and dense nuclei (Fig 3E, black arrows). Activated microglia with large rod-shaped nuclei surrounded degenerating neurons and formed clusters around small foci of degenerate/necrotic neurons (Fig 3F, green arrows).

No histological abnormalities were observed in the brain tissue of the mock mice (Fig 3G and 3H). Immunofluorescence staining of the cerebral cortex revealed that the PHEV nucleocapsid (N) protein was predominantly present in neurons, as identified by MAP2 staining (Fig 3I). Astrocytes, microglia, and oligodendroglia identified by GFAP, IBA1 and MBP staining, respectively, rarely showed costaining with the PHEV N protein (Fig 3J, 3K and 3L). These results suggest that PHEV is a neurotropic coronavirus that causes lethal disease resulting from brain infection and targets neurons in the CNS.

## PHEV hijacks the olfactory pathway for CNS invasion

Brain and nose tissues from PHEV-infected 3w BALB/c mice were sagittally sectioned, and the viral distribution was examined by performing immunofluorescence staining. In the olfactory epithelium (OE) of the nasal cavity, viral antigens were detected as early as 1 dpi in the cell body of OSNs (Fig 4A), and the number of PHEV-positive cells further increased at 3 and 5 dpi (Fig 4B and 4C). At 1 dpi, PHEV-infected cells were not observed in the brain (Fig 4D and S1 Table). At 3 dpi, the viral antigens were detected mainly in the brainstem and olfactory-associated regions, including the olfactory bulb (OB) and piriform cortex (Fig 4E and S1 Table). PHEV-positive cells were detected in the entire brain at 5 dpi, including the cerebellar regions (Fig 4F and S1 Table). Our findings suggest that PHEV is able to cross the neural-mucosal interface in the OE, and that the site of initial infection is the OB in the CNS.

To further validate this hypothesis, the blood, OE, OB, cerebrum, cerebellum, brainstem, and spinal cord of mock- and PHEV-infected 3w mice were harvested and evaluated by qRT–PCR (Fig 5A–5F). Compared to mock mice (Fig 5A), viral RNA was initially detected in the OE at 1 dpi (Fig 5B), followed by detection in the OB at 2 dpi (Fig 5C), suggesting that the olfactory route may be hijacked by PHEV for rapid neuroinvasion. Comparatively, the cerebrum and brainstem were positive for viral RNA at 3 dpi (Fig 5D). Viral RNA was globally distributed in the whole brain, including the cerebellum, at 4 dpi (Fig 5E) and spread to the spinal cord at 5 dpi (Fig 5F). No viral RNA was detected in blood samples from 1 to 5 dpi (Fig 5A–5F), excluding systemic spread via the blood and access to the CNS via transport across the blood–brain barrier or blood–cerebrospinal fluid barrier. Moreover, the viral antigen was detected in the olfactory nerves that connected the OE and OB (Fig 5G). Therefore, we concluded that PHEV invade the CNS of mice mainly through the olfactory pathway.

## The trigeminal nerve is an alternative route for PHEV neuroinvasion

To verify whether there are other neuroinvasion routes other than the olfactory nerve, we chemically destroyed the olfactory nerve endings of the OE by exposing 3w BALB/c mice to zinc sulfate ($ZnSO_4$) or Triton X-100 using previously described methods [35,36]. $ZnSO_4$ destroyed the OE by inducing rapid and selective OSN necrosis [37–39]. Mice in the PBS +PHEV group all died at 6 dpi, but 100% mortality was observed in $ZnSO_4$- and Triton X-100-treated PHEV-infected mice at 11 and 10 dpi, respectively (Fig 6A). This suggested that the chemical treatment delayed the time to death of mice but did not ultimately reduce mortality. We further detected the distribution of the viral antigens in the CNS of 3w mice treated with $ZnSO_4$ for 3 days prior to intranasal PHEV inoculation. Blood, OE, OB, cerebrum, cerebellum, brain stem, and spinal cord were harvested from different mice daily for 7 days and the presence of the viral RNA was detected in these samples by qRT–PCR. Notably, $ZnSO_4$-treated mice were resistant to PHEV neuroinvasion within 4 dpi (Fig 6B). The viral RNA was first detected in the brainstem at 5 dpi (Fig 6C) and throughout the whole brain at 7 dpi (Fig 6D). The data suggested that OE damage impeded PHEV entrance into the CNS via the olfactory nerve, but the olfactory route is not strictly the sole route for PHEV neuroinvasion. We

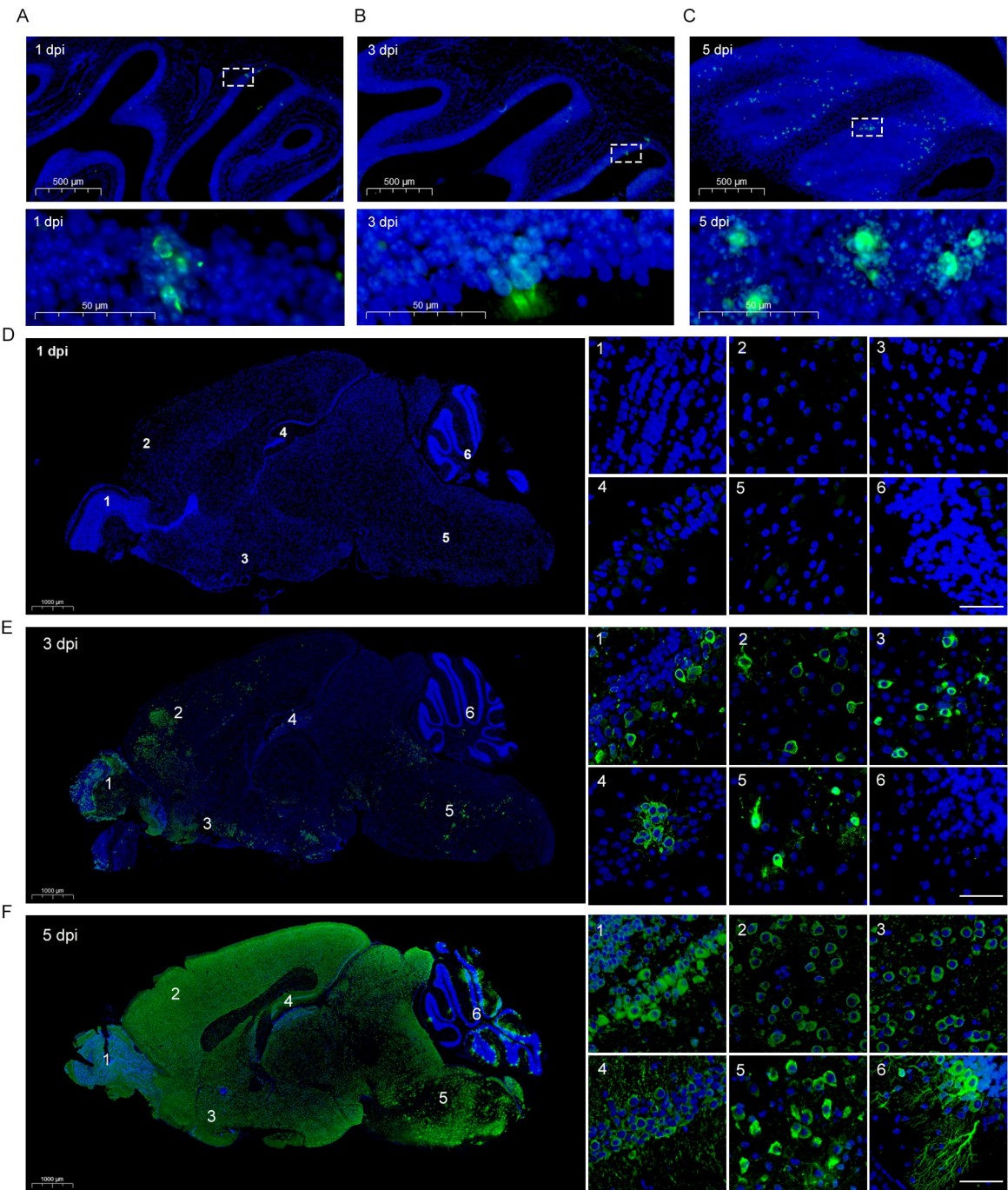

**Fig 4. Viral antigen distribution in the mouse OE and brain during PHEV infection.** The 3w BALB/c mice were inoculated intranasally with $10^{3.96}$ TCID$_{50}$ PHEV and sacrificed at 1, 3 and 5 dpi, respectively. **(A-C)** Representative images of immunofluorescence staining for viral antigens in the nasal cavity at 1 dpi (A), 3 dpi (B) and 5 dpi (C). The bottom panels show the magnified images of the dashed rectangles. Scale bars, 500 μm (A-C, top panels), 50 μm (A-C, bottom panels). **(D-F)** Representative images of immunofluorescence staining for viral antigens in the brains at 1 dpi (**D**), 3 dpi (**E**) and 5 dpi (**F**). Right panels represent the OB (panel **1**), cerebral cortex (panel **2**), piriform cortex (panel **3**), hippocampus (panel **4**), brain stem (panel **5**), and cerebellum (panel **6**). Scale bars, 1,000 μm (**D-F**, left panels), 30 μm (**D-F**, right panels). PHEV-N (green), DAPI (blue).

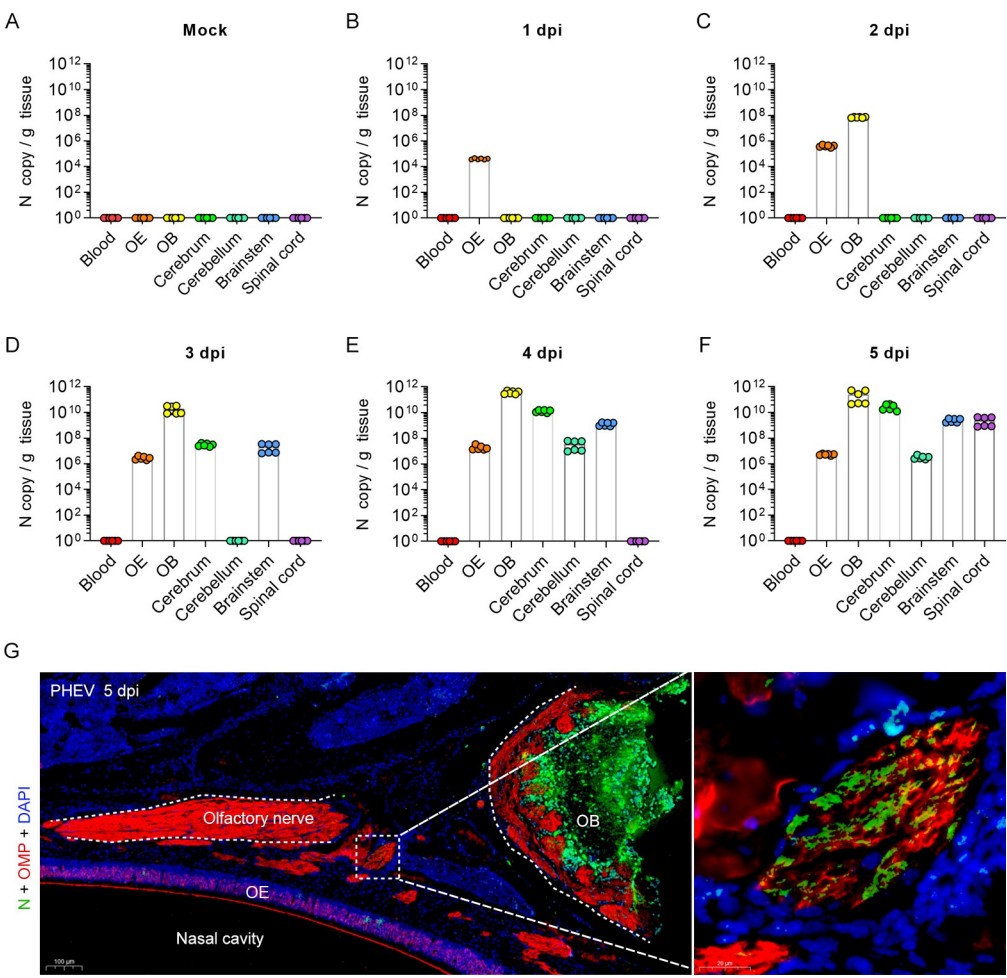

**Fig 5. PHEV invasion into the CNS via the olfactory nerve.** The 3w BALB/c mice were inoculated intranasally with $10^{3.96}$ TCID$_{50}$ PHEV and sacrificed at 1–5 dpi. **(A-F)** Mock and PHEV-infected CNS tissues (including OB, cerebrum, cerebellum, brain stem, and spinal cord), OE, and blood were collected for PHEV N RNA quantification by qRT–PCR (n = 6). The Y axis represents the PHEV N RNA copy number per gram of tissue. **(G)** Detection of PHEV-positive signals (N, green) in the olfactory nerve (OMP, red) of PHEV-infected 3w mice at 5 dpi. Nuclei stained with DAPI (blue). N, PHEV nucleocapsid protein; OMP, olfactory marker protein; OE, olfactory epithelium; OB, olfactory bulb. Scale bars, 100 μm (left panel), 20 μm (right panel).

then asked whether PHEV infects the subepithelial trigeminal nerve endings in the respiratory epithelium (RE) of the nose. The detection of viral antigens in the trigeminal nerve and trigeminal ganglion of 3w BALB/c mice provided a direct evidence of trigeminal nerve transmission of PHEV (Fig 7A and 7B). We concluded that in addition to the olfactory route, the subepithelial trigeminal nerve in the RE of the nasal cavity might represent an alternative route for PHEV neuroinvasion.

## Axonal transport enables PHEV neural transmission in the CNS

The sequential nature of PHEV distribution in the brain and spinal cord suggests a nonstochastic pathway of neural transmission within the CNS. Three-week BALB/c mice were infected intranasally with PHEV and brains were collected at 5 dpi. Immunofluorescence staining revealed that the N protein of PHEV colocalized with axons, as defined by staining for the marker βIII-tubulin, in several regions of the brain, such as the cerebral cortex and

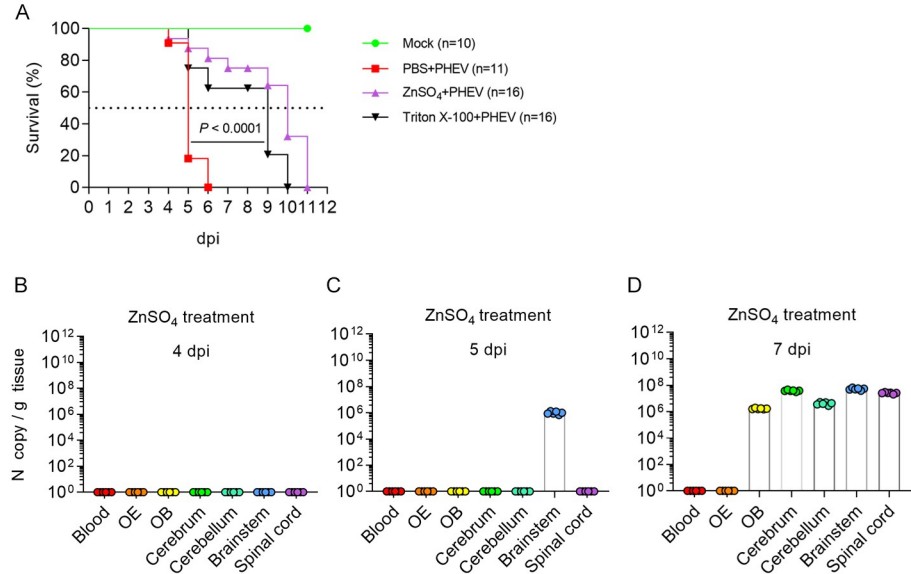

**Fig 6. Chemical treatment delayed the time of animal death. (A)** The 3w BALB/c mice were untreated (n = 10) or intranasally irrigated with 10 μl of PBS (n = 11), ZnSO₄ (0.17 M) (n = 16), or a 0.7% Triton X-100 (n = 16) solution in both nostrils daily for 3 days before intranasal inoculation with $10^{3.96}$ TCID$_{50}$ PHEV. The survival curves were plotted for these four groups. **(B-D)** The 3w BALB/c mice were intranasally irrigated with 10 μl of ZnSO₄ (0.17 M) in both nostrils 3 days before intranasal inoculation with $10^{3.96}$ TCID$_{50}$ PHEV. Mice were euthanized at different time points, and infected CNS tissues, OE, and blood were collected for PHEV N RNA quantification by qRT–PCR. Six mice per time point were analyzed. The Y axis represents the PHEV N RNA copy number per gram of tissue.

hippocampus (Fig 8A and 8B), suggesting that PHEV may be transported along axons. The localization of PHEV particles and axons was investigated in infected brains using transmission electron microscopy (TEM) to obtain deeper insights into this axonal association. As expected, assembled viral particles were associated with axonal structures (Fig 8C). Collectively, these data suggest that axonal transport may contribute to rapid, transneuronal spread of PHEV from initial sites to connected areas within the CNS.

## PHEV attacks OSNs and causes inflammation in the nasal cavity

In the nasal cavity of infected mice, PHEV-positive cells were scattered in the RE (Fig 9A, panels 1 and 2) and OE (Fig 9A, panels 3 and 4) at 5 dpi. Histopathologically, the PHEV infected RE was largely incomplete, with increased inflammatory cell infiltration and accumulation of luminal cell debris (Fig 9B, panels 1 and 2). Compared with the RE, the integrity of the OE was not significantly damaged, but intraepithelial and submucosal infiltration was also observed in the indicated regions (Fig 9B, panels 3 and 4). These findings indicate that the OE and RE in the nasal mucosa of mice are highly susceptible to PHEV infection.

We subsequently characterized the cells that were attacked in the OE and found that PHEV N protein was localized in OMP-expressing OSNs (Fig 10A), suggesting that PHEV specifically infected olfactory neurons. Furthermore, neural cell adhesion molecule (NCAM), a potential cell interacting partner of PHEV [22,23], was detected in the mouse OE using immunofluorescence staining, and most NCAM-expressing cells were colocalized with OMP-expressing OSNs (Fig 10B). Direct infection and virus-induced inflammatory responses potentially both lead to tissue damage and olfactory dysfunction. We then tested the effect of PHEV infection on the OE, and significant induction of a proinflammatory environment was observed (Fig

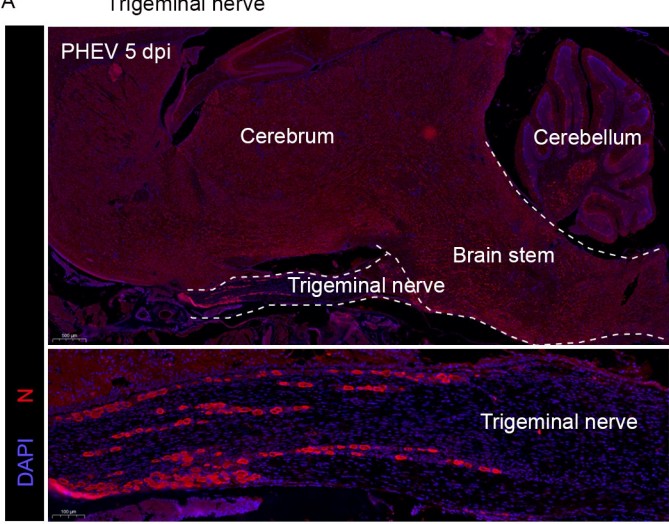

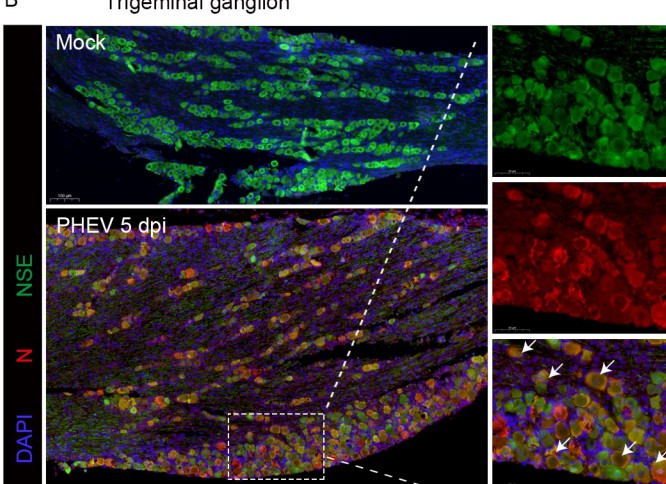

**Fig 7. The trigeminal nerve is an alternative route for PHEV neuroinvasion.** The 3w BALB/c mice were inoculated with $10^{3.96}$ TCID$_{50}$ PHEV by the intranasal route and sacrificed at 5 dpi. **(A)** Brains from PHEV-infected mice were subjected to immunofluorescence staining using a PHEV-N antibody (red). A magnified image of the trigeminal nerve is shown in the lower panel. **(B)** Trigeminal ganglions of mock- and PHEV-infected mice were immunofluorescence stained using NSE (green) and PHEV-N (red) antibodies. The right three panels represent magnified images of the area delimited by the dotted box. Arrows indicate cells colocalized with NSE and PHEV-N. DAPI stains nuclei (blue). NSE, neuron-specific enolase.

10C). Specifically, IL-1β levels were significantly increased as early as 1 dpi, while increases in IL-6 and CXCL10 mRNA levels were observed only at 3 and 5 dpi. CCL5, TNF-α, IFN-α, IFN-β, and IFN-γ were not significantly activated until 5 dpi compared with mock-infected mice (0 dpi). Consistently, immunohistochemical staining revealed a large number of IBA1-positive macrophages in the OE of PHEV-infected mice but few to no IBA1-positive cells in the OE of control mice (S4B and S4E and S4H Fig). Therefore, we demonstrated that PHEV exhibits strong tropism for OE and that CNS infection is associated with the inflammatory response of OE.

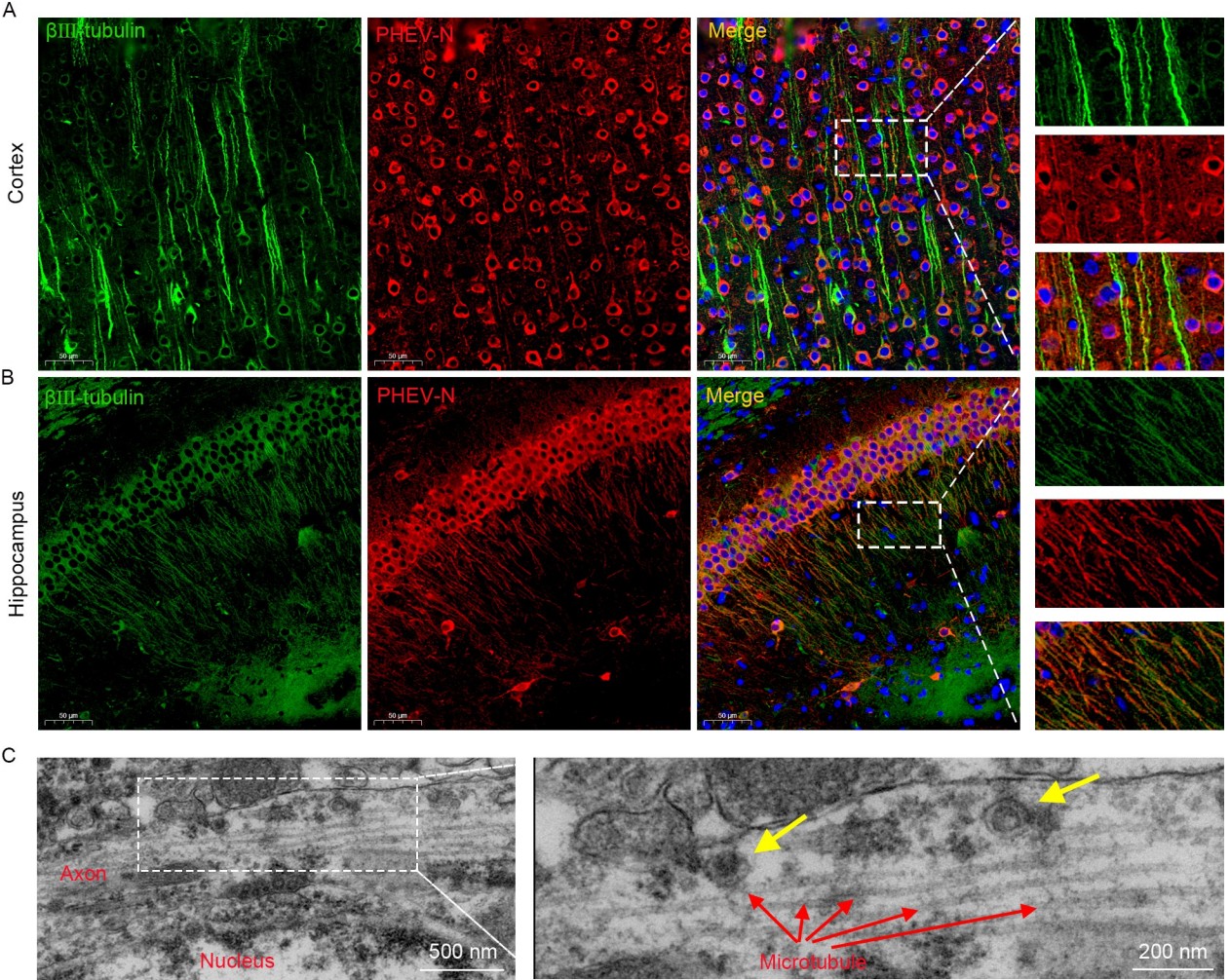

**Fig 8. PHEV N protein and viral particles are associated with axons in vivo.** The 3w BALB/c mice were intranasally inoculated with $10^{3.96}$ TCID$_{50}$ PHEV. **(A-B)** Brains were collected at 5 dpi, sagittally sectioned, and subjected to immunofluorescence staining with β III-tubulin (green) and PHEV-N protein (red) antibodies. The nuclei were stained with DAPI (blue). **(C)** Brains were collected at 5 dpi and analyzed by TEM. Red arrows, microtubules. Yellow arrows, virus particles.

## Dissemination of PHEV to the brain drives neuroinflammation

Because PHEV-infected mice exhibit olfactory dysfunction and rapid anterograde PHEV neuroinvasion via the olfactory route, we examined the viral distribution and the inflammatory response in the OB at 5 dpi. PHEV invades many regions in the OB, including the glomerular layer (GL), external plexiform layer (EPL), mitral cell layer (MCL), internal plexiform layer (IPL), and granular cell layer (GCL) (Fig 11A). Furthermore, the number of IBA1-positive macrophages/microglial cells increased significantly in the OB of PHEV-infected mice (S4C and S4F and S4I Fig). The high viral loads in the OB, together with macrophage/microglial cell infiltration, suggest that PHEV infection induces a bulbar inflammatory response.

Next, an RNA sequencing (RNA-seq) analysis was conducted to elucidate the characteristics of differentially expressed genes in the OB of PHEV-infected mice. An analysis of enriched Gene Ontology (GO) terms indicated that the 3542 upregulated genes were mainly associated with immune and inflammatory responses, including innate immunity (Toll-like and NOD-like receptor signaling pathway, NF-κB and Jak-STAT signaling pathway, cytokine–cytokine

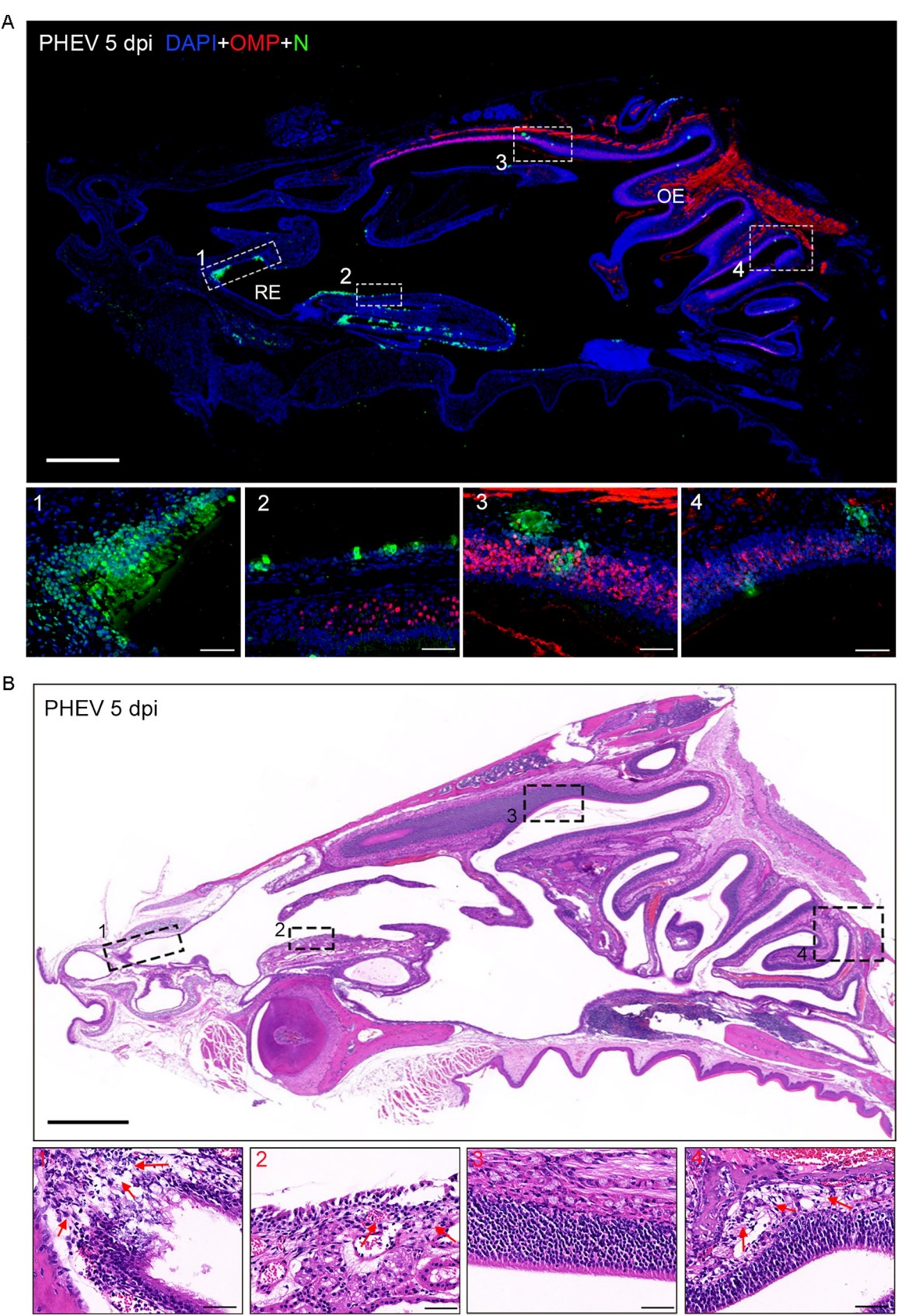

**Fig 9. Viral antigen distribution and histological changes in the nasal cavity.** The 3w BALB/c mice were intranasally inoculated with $10^{3.96}$ TCID$_{50}$ PHEV, and nose tissue samples were collected at 5 dpi. **(A)** Immunofluorescence staining of PHEV-N antigens in the RE and OE of the nasal cavity at 5 dpi. Magnified images indicate N-expressing cells in RE (panels **1–2**) and OE (panels **3–4**). OMP, olfactory marker protein (red); N, PHEV nucleocapsid protein (green). The nuclei were stained with DAPI (blue). Scale bars, 1,000 μm (upper panel), 50 μm (lower panels **1–4**). **(B)** H&E staining of tissue sections showed destruction and inflammatory cell infiltration of the RE (panels **1–2**) and OE (panels 3–4) at 5 dpi. The red arrows represent infiltrating inflammatory cells. Scale bars, 1,000 μm (upper panel), 50 μm (lower panels **1–4**).

interaction, complement, and coagulation cascades) and adaptive immunity (antigen processing and presentation, T and B cell receptor signaling pathway, and natural killer cell-mediated cytotoxicity) (Figs 11B and S5). The 1174 downregulated genes were mainly involved in nervous system functions, including neuronal development and differentiation, synaptic signaling, neurotransmitter transport, and anterograde transsynaptic signaling (Figs 11C and S5). These data indicate that PHEV infection induces a neuroinflammatory response and neuronal dysfunction in the OB, which may be responsible for anosmia.

## Discussion

Olfactory impairment and/or neurological manifestation are common symptoms of coronavirus diseases, but the exact mechanisms of neurological and olfactory dysfunction have not yet been clarified [40–43]. PHEV, a typical neurotropic coronavirus, is the causative agent of CNS disease in suckling pigs. Here, we performed virological, behavioral, and molecular studies in a PHEV-infected BALB/c mouse model that replicates olfactory, taste, and neurological dysfunction in coronavirus-related disease, providing a potential in vivo platform for investigating viral pathogenesis.

The nasal epithelium is the primary site for the neuroinvasiveness of most neurotropic respiratory viruses [8,44–48]. It consists of the RE and OE, which are located in the inferior-anterior and superior-posterior regions of the nasal cavity, respectively. Many OSNs in the OE comprise the olfactory nerves, and olfactory nerves coalesce to form larger nerve bundles that traverse the bony cribriform plate and terminate in the OB [49]. The olfactory nerve thus provides a direct pathway from the nasal cavity to the CNS [50–53]. In our model, we found that OE and RE are two areas infected by PHEV in the nasal cavity, with OSNs representing the major target cells in OE. The initial site of PHEV colonization in the brain is the OB and brain stem, followed by global transmission through the CNS, indicating that the olfactory nerve may be hijacked by PHEV for rapid CNS invasion. The essential function of OE during PHEV neuroinvasion was also confirmed, as chemically mediated degeneration of OSNs blocked PHEV access to the CNS and significantly improved the survival rate in mice in the early stage of infection. In addition, the ophthalmic branch of the trigeminal nerve, a sensory nerve that senses tactile stimuli, pain, and temperature, also innervates the nasal mucosa in the RE [54]. As shown in Fig 7, detection of the viral antigen in the trigeminal nerve and trigeminal ganglion provides direct evidence of trigeminal nerve transmission of PHEV. Thus, we conclude PHEV invades the CNS via the trigeminal nerve by infecting subepithelial nerve endings in the RE of the nasal cavity. Three main cell types and some glands are present in the RE, including ciliated cells, goblet cells, basal cells, serous glands, seromucous glands, and intraepithelial glands [55,56]. Nasal secretions and mucus produced by seromucous glands and goblet cells provide a physical barrier for the host against invading pathogens. This barrier may lead to less efficient PHEV infection of the subepithelial trigeminal nerve compared to the olfactory nerve. Interestingly, olfactory transmucosal invasion is also a port of CNS entry for SARS-CoV-2 and other neurotropic viruses [45]. SARS-CoV-2 was found in Neuropilin-1 (NRP-1) -positive

**Fig 10. Cell target and inflammatory responses in the OE of PHEV-infected mice.** The 3w BALB/c mice were intranasally inoculated with $10^{3.96}$ TCID$_{50}$ PHEV, and nose samples were collected at indicated times. **(A)** Immunofluorescence staining of PHEV-N (green) and olfactory neuron marker OMP (red) in OE at 5 dpi. Magnified images (panels **1–4**) show the colocalization of N with OMP-expressing OSNs. **(B)** Immunofluorescence staining of NCAM (green) and olfactory neuron marker OMP (red) in the mice OE at 5 dpi. NCAM, neural cell adhesion molecule. **(C)** Expression of IL-1β, IL-6, CXCL10, CCL5, TNF-α, IFN-α, IFN-β, and IFN-γ mRNA in homogenized nasal turbinate tissues was determined by qRT–PCR. Three mice per time point were analyzed. Data are normalized to GAPDH and presented as fold changes in expression relative to mock (0 dpi) and shown as the means ± SD. *P* values were determined by one-way ANOVA.

OSNs of the OE, OB and olfactory tracts in COVID-19 patients, indicating the critical roles of NPR1 in virus entry of the OSNs and anosmia [57–59]. Further research is needed to clarify if NRP1 serves as a host factor for PHEV infection.

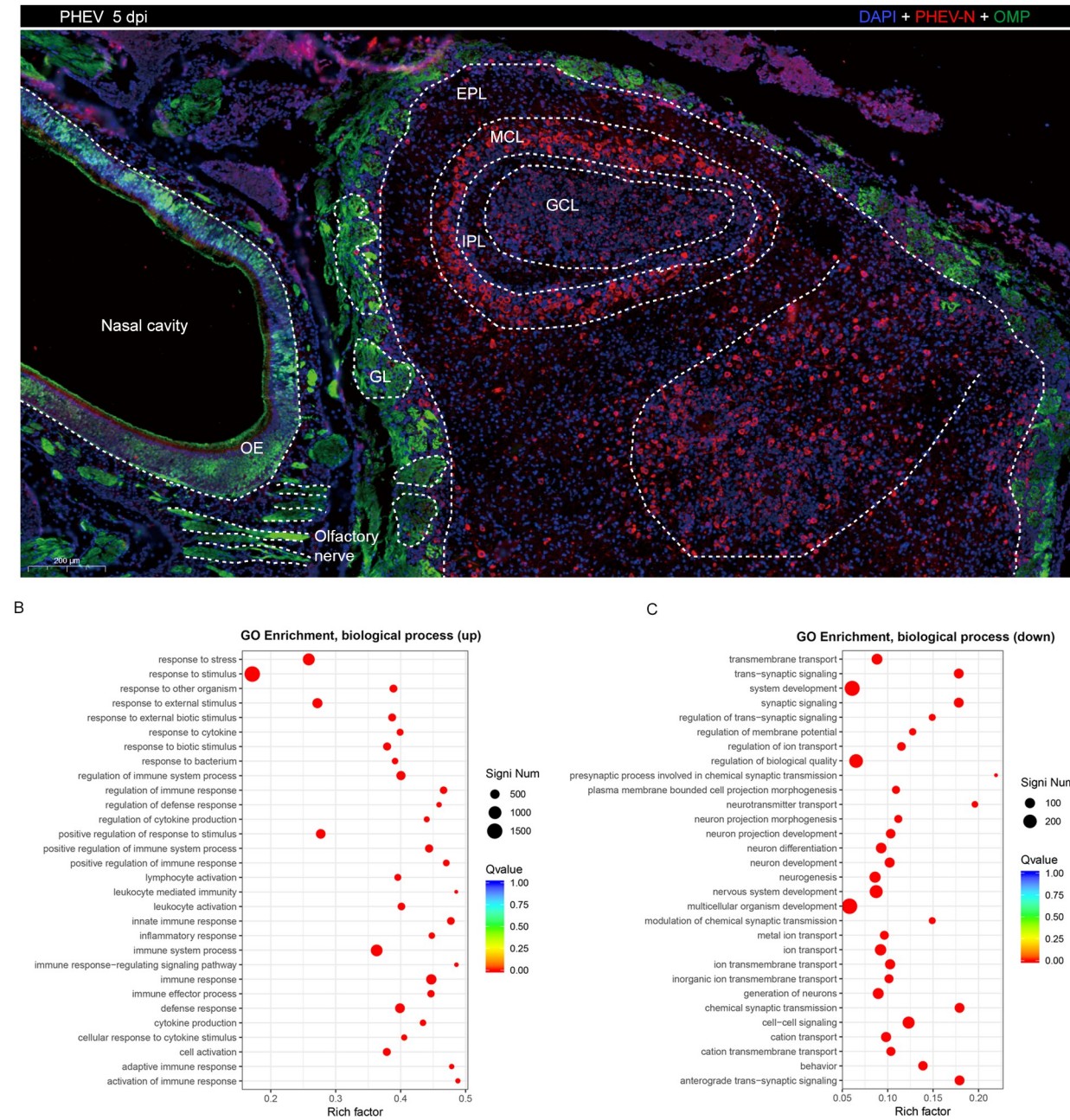

**Fig 11. Differentially expressed genes screened in the OB of PHEV-infected mice.** The 3w BALB/c mice were inoculated intranasally with $10^{3.96}$ TCID$_{50}$ PHEV. **(A)** At 5 dpi, mice were euthanized and OB samples were collected for immunofluorescence staining with antibodies against OMP (green) and PHEV-N (red). PHEV-positive cells were widely distributed in the OB, especially in the MCL and GCL. GL, glomerular layer; EPL, external plexiform layer; MCL, mitral cell layer; IPL, internal plexiform layer; GCL, granular cell layer. **(B-C)** Differentially expressed genes in the OB of PHEV-infected mice at 5 dpi were analyzed by RNA-seq. Scatter plot of biological processes identified in the GO enrichment analysis. The vertical axis represents the functional annotation information, and the horizontal axis represents the Rich factor corresponding to the function. (Only the 30 most enriched KEGG and GO terms are plotted for the significantly differentially expressed gene set).

Intranasal inoculation of PHEV resulted in anosmia and ageusia and induced an inflammatory response in the nasal cavity of mice, which characterized by the presence of IBA1-positive macrophages in the nasal epithelium and elevated levels of inflammatory cytokines. We

presume that the olfactory dysfunction may be caused by direct OSN infection and virus-induced inflammation. This hypothesis was supported by a report showing that SARS-CoV-2 infects OSNs in the OE of COVID-19 patients presenting with acute loss of smell, and viral replication in the OE is related to local inflammation [10]. SARS-CoV-2 also causes acute anosmia in golden Syrian hamsters, which lasted as long as the virus remained in the OE and OB with an enhanced immune response [10,60]. In the S2 Table, we showed how the PHEV-infected model and the SARS-CoV-2-infected models relate to findings identified in humans vis-a-vis anosmia and infection of the OE, RE, and OB. Furthermore, disruption of the cilial architecture in the OE or loss of cilia in the OSNs may also contribute to olfactory dysfunction [61]. Further studies should be performed to examine any correlations between the deciliation of the OE and olfactory dysfunction in PHEV-infected mice.

In this paper, we developed a novel neurological and olfactory dysfunction model based on intranasal inoculation of BALB/c mice with PHEV, and demonstrated its rapid CNS entry via the olfactory nerve in OE and/or the trigeminal nerve in RE. Similar to SARS-CoV-2, this prototype shows anosmia, ageusia, and neurological disorders along with productive replication of virus, robust inflammation, and tissue disorganization in the OE and CNS. Given that the high pathogenicity of human coronaviruses, relevant experiments must be conducted in a BSL-3 laboratory, which has undoubtedly hindered research progress. Therefore, the PHEV-infected BALB/c mice represent an excellent animal model for studying the viral pathogenesis, neuroinvasiveness, and neurovirulence of human pathogenic coronaviruses, albeit with limitations. For example, there is a large difference in the ability of neuroinvasiveness and neurotropism between SARS-CoV-2 and PHEV. SARS-CoV-2 does not replicate efficiently in the neurons and spread throughout the CNS causing a lethal encephalitis [62–64], although SARS-CoV-2 antigen is occasionally detected in the CNS of humans and relevant animal models [45,65]. In contrast, PHEV is strongly neurotropic but has little or no lung damage, leading to the restriction of using this model to evaluate pathogenic coronavirus-induced lung injury. Other limitation of experimental PHEV infection in mice as a model for SARS-CoV-2 associated CNS complications is the difference in host cell receptor usage between SARS-CoV-2 and PHEV. Angiotensin receptor 2 (ACE2) is a critical entry receptor for SARS-CoV-2 and it expresses diffusely on the mucous membrane of the entire oral cavity [66], but PHEV does not appear to use ACE2 as an entry receptor. Notably, both SARS-CoV-2 and PHEV could bind to the sialic acid (SA) of host cells [24,67,68]. Sialic acid is an essential element in the salivary mucin and may protect glycoproteins that transduce gustatory signals inside taste pores from enzymatic destruction [69,70]. PHEV may occupy sialic acid binding sites on taste buds, leading to gustatory particle destruction, but this remains to be elucidated in the future.

## Materials and methods

### Ethics statement

All experiments involving animals were approved by the Institutional Animal Care and Use Committee of the College of Veterinary Medicine, Jilin University, China (permission number KT202108025). All applicable institutional and/or national guidelines for the care and use of animals were followed.

### Study design

The main objective of this study was to investigate PHEV neuroinvasion in BALB/c mice and its potential use as a promising model to clarify coronavirus-associated neurological and olfactory and taste dysfunction. For assessments olfactory and taste functions, 6w BALB/c mice were used in behavioral experiments according to previously published protocols [34,71].

Intranasal PHEV inoculation has been successfully used in mice to mimic aerosol droplet infection in pigs [30,72]. We used 3w BALB/c mice to test neuroinvasion, which resulted in improved efficiency of intranasal inoculation [73].

## Cells and virus

Mouse neuroblastoma cells (Neuro-2a; ATCC CCL-131, Manassas, VA) were maintained in Dulbecco's modified Eagle's medium (DMEM; Meilunbio, Dalian, CN) supplemented with 6% fetal bovine serum (FBS; BI, Kibbutz Beit Haemek, Israel), penicillin and streptomycin (100 U/ml and 100 μg/ml, respectively) and incubated at 37°C with 5% $CO_2$. The PHEV CC14 strain (GenBank accession number: MF083115) was isolated from naturally infected piglets with neurological symptoms, vomiting, diarrhea, and wasting, and was propagated in Neuro-2a cells [74,75].

## Antibodies

Rabbit polyclonal anti-olfactory marker protein antibody (ab183947), rabbit polyclonal anti-MAP2 antibody (ab21693), rabbit polyclonal anti-GFAP antibody (ab207165), rabbit polyclonal anti-IBA1 antibody (ab178846) and rabbit polyclonal anti-MBP antibody (ab218011) were obtained from Abcam (Cambridge, MA). The rabbit polyclonal anti-NSE antibody was obtained from Servicebio (Wuhan, CN). The mouse monoclonal anti-β3-tubulin (TU-20) antibody (#4466), Alexa Fluor 488-conjugated goat anti-rabbit antibody (#4412), Alexa Fluor 488-conjugated goat anti-mouse antibody (#4408), Alexa Fluor 594-conjugated goat anti-rabbit antibody (#8889) and Alexa Fluor 594-conjugated goat anti-mouse antibody (#8890) were purchased from Cell Signaling Technology (Beverly, MA).

Rabbit polyclonal anti-PHEV-nucleoprotein antibodies were prepared and stored in our laboratory. Briefly, the recombinant full-length PHEV-N protein was generated based on the sequence of the PHEV CC14 strain. First, the PHEV-N coding region was amplified using RT–PCR and cloned into the pET-32a(+) vector. Second, the recombinant protein was expressed and purified with His-tag Purification Resin (P2233, Beyotime). Third, the polyclonal antibody was prepared in New Zealand white rabbits by subcutaneously injecting 100 μg of recombinant protein combined with an equal volume of Freund's incomplete adjuvant at eight different sites. Rabbits were boosted three times at 2-week intervals.

## Animal experiments

Three-week-old (3w) and six-week-old (6w) BALB/c mice were purchased from Liaoning Changsheng Biotechnology Co., Ltd. (Shengyang, CN), provided ad libitum access to water and a standard chow diet, and maintained in the animal facility in the College of Veterinary Medicine, Jilin University. Mice were subjected to intranasal inoculation with $10^{3.96}$ $TCID_{50}$ of PHEV (strain CC14) diluted in 20 μl of phosphate-buffered saline (PBS, 0.01 M, pH 7.4) (10 μl were instilled dropwise in each nostril) or a sham inoculation with the same dose of sterile PBS under 1–3% isoflurane anesthesia. Following inoculation, mice were monitored twice daily and a clinical score was recorded based on an Institutional Animal Care and Use Committee (IACUC)-approved scoring system for a maximum score of 5, and the evaluation indicators included body weight, respiration, general appearance, responsiveness and neurological signs (Table 1) [76]. Humane endpoints were established based on the IACUC-approved clinical scoring system. Mice were considered moribund and humanely euthanized when the cumulative clinical score of 4 or weight loss greater than 20% was observed. Mice were euthanized when they reached humane endpoints. Briefly, mice were exposed to 5% isoflurane for 5 min

**Table 1. Clinical scoring system used for PHEV-infected mice.**

| Category | Score = Criteria |
|---|---|
| Body weight | 1 = 10–19% of body weight loss |
| Respiration | 1 = rapid, shallow, polypnea |
| Appearance | 1 = ruffled fur, hunched posture |
| Responsiveness | 1 = low to moderate movement, paralysis |
| Neurologic signs | 1 = tremors, ataxia, circling, and seizures |

in a plexiglass chamber before being decapitated when fully sedated, as determined by the absence of an active paw reflex.

For survival curve experiments, any mice that reached the humane endpoints were humanely euthanized immediately, regardless of the day. For time course experiments, a subset of mice were humanely euthanized and tissue samples were collected at 1, 2, 3, 4, and 5 dpi. Blood was collected through eyeball extraction under deep anesthesia with isoflurane, and serum was obtained by centrifugation at 4˚C, 4,000 xg for 10 minutes. Samples of heart, liver, spleen, lung, kidney, small intestine, colon, brain and spinal cord were collected after transcardial perfusion with 4% (wt/vol) paraformaldehyde, and then stored at -80˚C for qRT-PCR analysis or fixed in 10% neutral buffered formalin solution for histology analysis.

## Sucrose preference test

Taste function in mice was assessed with a sucrose preference test as previously described [71]. Briefly, the 6w BALB/c mice (male and female) were subjected to 48 h of continuous exposure to both 1% sucrose water and regular water for adaptation. All mice had ad libitum access to laboratory chow. During the preference test after intranasal PHEV and PBS inoculation, mice were deprived of water for 6 h, and individual overnight (12 h) testing, which corresponds to the circadian rhythms of the drinking of mice, was performed. Abnormalities in taste function were indicated by the reduction in the sucrose preference ratio (preference = sucrose intake/ total intake × 100%) in PHEV-infected and control mice.

## Buried food finding test

The buried food finding test was performed as previously described with a few modifications [34]. Six-week BALB/c mice (male and female) were used only once per day for each test, and the food position was changed daily. Mice were fasted for 12 h and sensitized to food for 5 min before testing and then individually placed into a fresh cage with food pellets hidden below the bedding. The latency to locate and dig the buried food was recorded using a stopwatch. The experiment was carried out for a 3 min period, and if the mice could not find the food, the time was recorded as 3 min.

## Social scent discrimination tests

The social scent discrimination test was performed to evaluate the ability of mice to detect and differentiate different odors using previously described method, with a few modifications [9]. Briefly, two identical 2-ml Eppendorf tubes or 3-cm dishes were separately sealed with bedding from different mouse cages overnight and placed at two different corners in a fresh cage (28 cm × 24 cm × 17 cm). For male 6w mice, the bedding was collected from female mouse cages and male cages. For female 6w mice, the bedding was collected from the home cage ('familiar') and another cage ('novel'). Next, PHEV-infected mice or mock-infected mice were released into fresh cages. Time spent sniffing within a 3-min period was measured. Each mouse was

performed one trial each day with the position of tubes or dishes changed daily. To avoid the interference of decreased mobility or malaise, the preference index for both sexes was also calculated as previously reported [9].

## Chemical deafferentation of olfactory nerve

To destroy the OE and block olfactory viral neuroinvasion, the 3w BALB/c mice were intranasally irrigated with 20 μL of $ZnSO_4$ (0.17 M) or 0.7% Triton X-100 solution in both nostrils (10 μl were instilled dropwise in each nostril) daily for 3 days before intranasal PHEV inoculation. We investigated the effect of OE destruction on mouse survival by plotting survival curves using the aforementioned humane endpoint criteria. Mice were euthanized, and whole infected CNS tissues (including OB, cerebrum, cerebellum, brain stem, and spinal cord) and blood were collected from different mice daily until the fifth day post-infection for the qRT–PCR analysis.

## qRT–PCR

Total RNA was purified from 100 mg of homogenized tissues using a TransZol Up Plus RNA Kit (ER501-01, Transgen, Beijing, CN) according to the manufacturer's instructions. One microgram of total RNA was reverse transcribed to first-strand cDNAs using EasyScript All-in-One First-Strand cDNA Synthesis SuperMix for qPCR (AE341-02, Transgen). The primers used to measure the abundance of mouse cytokines and chemokines were previously reported [9]. The comparative $\Delta\Delta C_T$ method was used to calculate the relative abundance of transcripts using the average values of each gene and normalized to GAPDH. For the detection of viral RNA, cDNAs from mouse tissues were subjected to the amplification of genomic RNA for the PHEV N protein by qRT–PCR using the following primers: PHEV-N-F, 5′-TC TGGGAATCCTGACGAG-3′; PHEV-N-R, 5′-AGGCGCTGCAACACTTAC-3′. A standard curve was constructed for each PCR using $10^1$–$10^8$ copies of a PHEV-N plasmid to calculate copy numbers for each reaction.

## Histopathology, immunofluorescence (IF), and immunohistochemistry (IHC)

Mice were anesthetized and transcardially perfused with 4% (wt/vol) paraformaldehyde. Then, tissues were removed and fixed with a 10% neutral buffered formalin solution. For histopathology, paraffin-embedded tissues were sectioned routinely (3–5 μm thickness) and stained with hematoxylin and eosin (H&E). For IF, sections were deparaffinized, rehydrated, and antigen retrieval was performed using pH 6.0 citrate buffer (Servicebio, CN). Sections were incubated with blocking reagent (5% skim milk) followed by primary antibodies overnight at 4˚C and then incubated with the appropriate fluorescently labeled secondary antibody for 1 h at room temperature. All sections were then mounted using antifade mounting medium with DAPI (P0131, Beyotime) and scanned with a PANNORAMIC MIDI II automatic digital slide scanner (3DHISTECH, Budapest, Hungary). For IHC, the sections were deparaffinized, rehydrated, and subjected to antigen retrieval according to the instructions of the UltraSensitive SP (Mouse/Rabbit) IHC Kit.

## Viral titration

Briefly, tissue samples were weighed and homogenized in 1 ml of 2% DMEM, and tissue homogenate supernatants were serially diluted ten-fold in DMEM. Neuro-2a cells seeded in ninety-six-well plates were inoculated with serial dilutions of tissue homogenates at 37˚C with

5% $CO_2$ for 1 h. After removing the inocula, DMEM containing 2% FBS was added to the plates for 3 days. The $TCID_{50}$ was calculated by the method of Reed-Muench and presented as $TCID_{50}$/g of tissue weight.

## Transmission electron microscopy (TEM)

The TEM analysis was performed as previously described [32]. Briefly, PHEV-infected mouse brain samples were collected, fixed with 2.5% glutaraldehyde for 24 h, and then postfixed with 1% osmium tetroxide for 2 h. After dehydration using a graded ethanol series, the specimens were embedded in Renlam resin. The ultrathin sections were cut using a diamond knife, transferred onto slot grids and then stained with 2% uranyl acetate and 0.4% lead citrate. The ultrathin sections were visualized using a HITACHI HT7800 microscope.

## Transcriptome sequencing

Transcriptome sequencing and data analysis were performed at Sangon Biotech (Shanghai) Co., Ltd. Briefly, cDNA libraries were constructed according to the manufacturers' instructions (Hieff NGS MaxUp Dual-mode mRNA Library Prep Kit for Illumina) and sequenced by an Illumina HiSeq 2,500 sequencer. After quality control of the original sequencing data, reads were mapped to the reference genome sequence. Reads mapped to the genes were counted, and gene expression was calculated.

## Statistical analysis

Graphics and statistical tests were performed using GraphPad Prism v8.0 software (GraphPad Software, San Diego, CA). Data are presented as the means ± SD. Statistical significance was considered at $^*P<0.05$, $^{**}P<0.01$, $^{***}P<0.001$ and $^{****}P<0.0001$.

## Supporting information

**S1 Fig. Sex-independent clinical symptoms of mice after PHEV infection.** Male and female 3w and 6w BALB/c mice were intranasally inoculated with $10^{3.96}$ $TCID_{50}$ PHEV per mouse (n = 5 mice/age/sex). **(A)** Mortality of PHEV-infected female and male mice. **(B)** Percentage of initial body weight. **(C)** Clinical scores. $P$ values were calculated by log-rank (Mantel–Cox) tests (survival) (**A**), Wilcoxon matched-pairs rank test (B), and one-way ANOVA (**C**). Data are presented as the means ± SD.
(TIF)

**S2 Fig. Visible food finding test.** The 6w male and female BALB/c mice were intranasally inoculated with $10^{3.96}$ $TCID_{50}$ PHEV or mock-infected with PBS (M) before the visible food finding test. (A) Time spent by mock or infected mice (1–3 dpi) finding the visible food. The dashed line represents the time limit of 3 min. Each circle represents a mouse. (B) Percentage of mice that successfully found visible food within 3 min. For male mice (mock: n = 13, 1–3 dpi: n = 6), female mice (mock: n = 13, 1–3 dpi: n = 7). $P$ values were calculated by one-way ANOVA (**A**). Data are presented as the means ± SD.
(TIF)

**S3 Fig. Histopathological analysis of the liver, spleen, kidney, small intestine and colon in PHEV-infected mice.** The 3w BALB/c mice were euthanized at 0, 3 and 5 dpi after $10^{3.96}$ $TCID_{50}$ PHEV inoculation, and tissues were collected for the histological examination. The liver (**A**), spleen (**B**), kidney (**C**), small intestine (**D**), and colon (**E**) were analyzed. There were no substantial histopathological changes in the organs of PHEV-infected mice. Scale bars,

50 μm (**A**), 100 μm (**B**, **C** and **E**), 200 μm (**D**), H&E staining. Two sections of each organ from 3 mice per group were analyzed.
(TIF)

**S4 Fig. Inflammatory cell accumulation in the OE and OB of PHEV-infected mice.** The 3w BALB/c mice were inoculated intranasally with $10^{3.96}$ TCID$_{50}$ PHEV. OB and nose tissues were collected at 0, 3 and 5 dpi for IHC with an antibody against IBA1. Compared with control tissues (**A-C**), incremental accumulation of IBA1-positive macrophages/microglia cells was observed in the OE and OB at 3 (**D-F**) and 5 (**G-I**) dpi. Scale bars, 500 μm (**A, D, G**), 50 μm (**B, C, E, F, H, I**). Two sections of each tissue from 3 mice per group were analyzed, and representative images are shown.
(TIF)

**S5 Fig. Differentially expressed genes (DEGs) in the OB identified by transcriptomic sequencing (RNA-seq).** The 3w BALB/c mice were inoculated intranasally with $10^{3.96}$ TCID$_{50}$ PHEV and OB samples were collected at 5 dpi for RNA-seq. (**A**) Volcano plot. The horizontal axis represents the fold change in DEGs, and the vertical axis represents the Benjamini–Hochberg corrected *p value* on a logarithmic scale (-log10). Each dot represents a gene, where red dots represent up-regulated genes, green dots represent down-regulated genes, and black dots represent non-differentially expressed genes. **b**, Cluster heatmap of DEGs. Each row represents a gene, and each column represents a sample. The color represents the expression level of the gene, the red color represents a high expression level, and the green color represents a low expression level. **c**, Scatter plot of the KEGG pathway enrichment analysis. P, PHEV-infected sample; M, mock sample.
(TIF)

**S1 Table. Distribution of viral antigen in the brains of PHEV-infected mice.**
(DOCX)

**S2 Table. Comparison of anosmia and brain infection among the PHEV-infected mouse model, SARS-CoV-2-infected hamster model, SARS-CoV-2-infected humanized ACE2 mouse model and deceased COVID-19 patients.**
(DOCX)

## Acknowledgments

The authors thank Dr. Chen-Hsuan Liu from the School of Veterinary Medicine, National Taiwan University for his assistance with the histopathological analysis.

## Author Contributions

**Conceptualization:** Junchao Shi, Zi Li, Wenqi He.

**Formal analysis:** Yungang Lan, Kui Zhao, Wenqi He.

**Funding acquisition:** Zi Li, Feng Gao, Wenqi He.

**Investigation:** Junchao Shi, Jing Zhang, Rongyi Xu, Zhenzhen Wang.

**Methodology:** Junchao Shi, Jiyu Guan, Rui Gao.

**Resources:** Huijun Lu, Baofeng Xu.

**Supervision:** Feng Gao, Wenqi He.

**Writing – original draft:** Junchao Shi, Zi Li.

**Writing – review & editing:** Kui Zhao, Feng Gao, Wenqi He.

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
