## [Decision Letter · Decision Letter 0]

17 Mar 2022

Dear Dr. He,

Thank you very much for submitting your manuscript "PHEV infection: a promising model of betacoronavirus-associated neurological and olfactory dysfunction" (PPATHOGENS-D-22-00231) for review by PLOS PATHOGENS. Your manuscript was fully evaluated at the editorial level and by independent peer reviewers. The reviewers appreciated the attention to an important problem, but raised some substantial concerns about the manuscript as it currently stands. These issues must be addressed before we would be willing to consider a revised version of your study. We cannot, of course, promise publication at that time. We therefore ask you to modify the manuscript according to the review recommendations before we can consider your manuscript for acceptance. Your revisions should address the specific points made by each reviewer.

We are returning your manuscript with three reviews. All Reviewers acknowledged that the manuscript contains a comprehensive set of new and interesting data, but raised concerns towards the context of the study as it is currently presented and the imprecise description of the experimental design. Currently, the experimental design and the rationale behind this design is not always clear. This includes f.e. the rationale to initially perform experiments in 3 and 6 week old mice, and in the latter experiments only 3 week old mice (Reviewer 2) as well as the methodological choices and descriptions (reviewer 1 & 3). In addition, the number of infected mice should be clear for each set of data, as well as the time post inoculation for the presented data (this is unclear in f.e. Figure 2).

In addition, the current manuscript is written in the context of SARS-CoV-2, and how studying the pathogenesis of PHEV could provide insights into the pathogenesis of SARS-CoV-2 infection. However, the reviewers agree that extrapolation of the acquired data to SARS-CoV-2 infection is too far-fetched. There is no side-by-side comparison between PHEV and SARS-CoV-2 in the experimental design, nor in the discussion based on previously published papers. Despite some overlap in the route that both PHEV and SARS-CoV-2 can use to enter the CNS, the cell tropism and ability to spread throughout the CNS differs largely.

We cannot make any decision about publication until we have seen the revised manuscript and your response to the reviewers' comments. Your revised manuscript is also likely to be sent to reviewers for further evaluation.

Sincerely,

Debby van Riel

Guest Editor

PLOS Pathogens

Ron Fouchier

Section Editor

PLOS Pathogens

Kasturi Haldar

Editor-in-Chief

PLOS Pathogens

orcid.org/0000-0001-5065-158X

Michael Malim

Editor-in-Chief

PLOS Pathogens

orcid.org/0000-0002-7699-2064

Reviewer's Responses to Questions

**Part I - Summary**

Reviewer #1: In this manuscript by Shi et al, authors utilize the BALB/c mouse model to investigate the infectivity and tropism of a porcine betacoronavirus (PHEV) in neurological and olfactory-associated tissues. Authors conduct several behavioral studies to measure changes in CNS-related functions in infected mice, and extensively employ immunohistochemistry-based assessments to identify specific tissue structures post-inoculation that likely contribute to these in vivo readouts. They identify specific nerves targeted by PHEV infection and link viral replication at these sites with pathogenic responses. The study appears generally well-controlled, though there are places in the text that make it difficult to follow what exactly is being shown. The manuscript is generally well-referenced and logically organized, with a few exceptions. However, the authors heavily interpret their results in the context of SARS-CoV-2 infection, at the expense of focusing on what their findings mean in the context of PHEV infection. Overall there are several areas throughout the manuscript that warrant improvement.

Reviewer #2: In the manuscript by Shi et al the authors investigate the neuroinvasion routes for the betacornavirus called Porcine hemagglutingating encephalomyelitis virus (PHEV) using a BALB/c murine system. Their work provides mechanistic understanding regarding PHEV neuroinvasion and neurotropism. The main findings are that the virus can invade the CNS via the olfactory epithelium/olfactory nerve, in addition to via the trigeminal nerve. The findings are convincing and the studies are well executed. This work adds to our understanding of PHEV and its ability to invade the CNS. However, the authors do not discuss how this work relates to PHEV pathogenesis, despite this virus being an important agricultural pathogen and a major source of economic loss on many farms (Mora-Diaz, et al; Front Vet Sci. 2019 doi: 10.3389/fvets.2019.00053). Rather, these data are postulated to represent a low containment model to study SARS-CoV-2 neuropathogenesis. Here the authors also fall short because they do not adequately related their findings to relevant SARS-CoV-2 models, most importantly the hamster models that have provided insight into anosmia and brain infection that is observed in humans. The manuscript also needs a clear discussion pertaining to the limitations of the surrogate PHEV model as it relates to SARS-CoV-2.

Reviewer #3: The manuscript „PHEV infection: a promising model of betacoronavirus-associated neurological and olfactory disfunction” by Shi et al. is describing at mouse model to study the neuroinvasion and neurotropism about PHEV. The authors give a detailed description about pathogenicity and time course of infection and give a great overview of the mechanism of CNS infection of PHEV. Although this manuscript presents a comprehensive set of new data, the description of experiments and the interpretation of results is making it hard for the reader to follow the thoughts of the authors.

**Part II – Major Issues: Key Experiments Required for Acceptance**

Reviewer #1: -The majority of the introduction and discussion sections pertain to SARS-CoV-2 infection, not PHEV infection. It is understandable that this study with PHEV was undertaken to provide contextual information to SARS-CoV-2 infection, but without side-by-side data showing that SARS-CoV-2 and PHEV infection target similar tissues or elicit comparable responses, it’s difficult to use the PHEV model established here as a surrogate virus to directly inform SARS-CoV-2/COVID-19 studies. What receptor(s) does PHEV employ to infect CNS-related cells, what is their relative distribution on the specific CNS-associated tissues employed in this study, and are SARS-CoV-2 receptors present on the same tissues in permissive species? This is all critical information that appears to be missing or underemphasized within the text. While SARS-CoV-2 cannot infect BALB/c mice (limiting side-by-side assessments of both viruses in the same model), providing some scope of data regarding which mammalian models would be appropriate for employing any subsequent results obtained in the PHEV mouse model discussed here seems appropriate; without this context and additional information, the authors should temper their applicability of PHEV results to SARS-CoV-2 infection studies.

-The methods section is lacking the humane endpoints employed in this study (were mice euthanized once they reached specific endpoint cutoffs)? The survival curves presented are difficult to interpret without understanding which specific criteria (weight loss, lethargy, specific neurological symptoms, etc) necessitated this euthanasia. It is also unclear as currently written if the mice that succumbed to infection in Fig 4A (with delays in death relative to Figure 1B) were due to the same manifestation of endpoint criteria or if the prolonged course of infection resulted in additional sequalae to develop. The method of humane euthanasia also needs to be disclosed (chemical, physical, etc).

-As described in the text, Figure 1 purports to show differences between mock-infected and PHEV-infected mice, but many of the display images (1D-O) only appear to show breakdowns between male and female mice, and not infected vs mock. Figure legend for Fig 1 also states that bars represent mean + SD, but SD does not appear to be included in panels E-F, making it difficult for the reader to understand if the bars represent a different parameter or not.

Reviewer #2: 1. While the authors cite the hamster model papers that examined SARS-CoV-2 in the olfactory epithelium and brain infection (de Melo at al), there is virtually no discussion about the hamster SARS-CoV-2 model or these cited studies. The hamster system is extensively used for SARS-COV-2 research, including infection in the olfactory cavity. I think this should be at least a paragraph in the discussion addressing the hamster models (and other models) and a comparison to their results. Ideally, the authors should consider a table showing how the PHEV BALB/c model and the hamster models relate to what has been identified in humans vis-a-vis anosmia and infection of the olfactory bulb.

2. There appears to be no lung involvement in the PHEV/BALB/c system. Some evidence suggests that SARS-COV-2 can usurp the vagus nerve subsequent to lung infection to invade the CNS (Bulfamante, G et al J. Neurol; 10.1007/s00415-021-10604-8). Does a lack of respiratory involvement impair the in the PHEV system as a surrogate for SARS-CoV-2 neuroinvasion? This should be considered in a limitations paragraph along with other pros and cons of the PHEV surrogate model.

3. The introduction and discussion should clearly and conspicuously address the agricultural disease caused by PHEV, including the fact that it is an important pathogen. How the author’s data relates to what is known about this disease should also be added to the discussion in its own paragraph. The authors may want to consider reformatting the manuscript as a PHEV pathogenesis paper, then mention its potential use as a surrogate SARS-COV-2 system in the dissuasion.

4. The discussion mentions the potential involvement of ACE2 binding as being important for loss of taste, etc. However, I don’t believe that PHEV binds ACE2, rather it binds sialic acid and potentially the Neural Cell Adhesion Molecule. If binding to ACE2 were critical for SARS-CoV-2 neuroinvasion, but PHEV didn’t use this receptor, it would make this a poor model to mimic COVID-19. The authors need to explicitly state the receptor that PHEV uses and then modify the discussion as to how this may be a limitation to this model.

5. It isn’t clear why the authors used 3 and 6 week old mice initially, then changed over to just 3 week old mice for subsequent studies. 6 week old mice would be preferred as these are adult mice. Is there a reason to only have used 3 week old mice?

Reviewer #3: - I am missing a consistency of methodological workup of samples in terms of titration and PCR. For some samples, titrations are performed to quantify infectious virus, for some not. Either explain your workflow or keep it consistent throughout all experiments

- It is very hard to follow which animals (gender, age) and how many (n=?) were used for the different experiments. It seems as if a lot of different sub-experiments were performed with partially different conditions. It would be ideal for reader to follow the workflow and thoughts of the authors if a graphical experimental setup or a table would give an overview of those aspects.

**Part III – Minor Issues: Editorial and Data Presentation Modifications**

Reviewer #1: -Abstract and author summary require editing for English prose (furthermore, use of “biosafety” and “biosafe” in the abstract do not make logical sense as current written). There are also some passages within the main text where phrasing is too colloquial (e.g. line 100, “attractive”, line 152 “suffering mice”).

-anosmia and ageusia should be better defined in the author summary as this is intended for a more general audience.

-lines 127-8, what additional “neurological complications” are being alluded to outside of anosmia and ageusia?

-Authors should be commended for employing both male and female mice of two ages in this study, however, it is unclear why authors elected to use the 3 week vs the 6 week old mice in different studies. Was there a research purposes for choosing one age over another in selected studies (e.g. studies shown in Figure 3, line 147) or were findings interchangeable? If there was a reason, the authors should better disclose this for clarity. There are also places the age of mice employed is not clearly stated (e.g. Figure 5 legend)

-Figure 4, it is unclear if the data presented is from ZnSO4-treated mice or not (e.g. Figure 4B, the text implies that the treated mice are being shown, but this is not specified in the legend text, and then panels 4C-F seem to imply that what is being shown is from non-treated mice). If ZnSO4-treated mice are being shown, showing both treated- and untreated- mice here would seem appropriate so that the reader can better appreciate differences that are specifically attributed to the destroyed olfactory nerve endings induced by ZnSO4.

-Figure 5D, what is fold induction relative to? Uninfected mice? The methods, figure legend, or text do not specify this. It would be beneficial if baseline quantities of each analyte examined were disclosed in the legend or methods so that the reader could better understand the relative expression (and magnitude of change when appropriate) for each.

Reviewer #2: 1. line 26, “a biosafety betaconronavirus” does not make sense. This should be reworded. Maybe “non-human pathogenic..”?

2. Line 41-43. “...utilized as a protoype to recapitulate neurological and olfactory dysfunction in a subset of COVID-19 involvement under...” should be reworded. Perhaps “....olfactory dysfunction observed in human SARS-COV-2 infections?

3. Line 106. The strain of virus used should be added to be consistent (CC14?).

4. The method to disrupt the olfactory epithelium via TritonX and ZnSO4 needs to be explicitly in the methods section, not merely a reference. One of the papers referenced, also neglected to put the method in the paper and only referenced another paper.

5. Line 249. “absence of a decreased viral load”. It is unclear what they are trying to convey with this statement. Please clarify.

6. The paper should reference Reyna, RA et al “Recovery of anosmia in hamsters infected with SARS-CoV-2 is correlated with repair of the olfactory epithelium” Scientific reports doi.org.10/1038/S41598-021-04622-9. The findings should incorporated into the discussion.

7. The authors should look at a recent Trends in Neuroscience paper “The neuroinvasiveness, neurotropism and neurovirulence of SARS-CoV-2”. This may help as they revamp the discussion to talk about the hamster models and how these rodent systems relate to neuroinvasiveness of SARS-CoV-2. doi.org/10.1016/j.tins.2022.02.006.

Reviewer #3: - Revise English language! Especially the abstract and author summary need to be revised by a native speaker. Those two paragraphs are nearly impossible to understand due discrepancies in English language. It is getting better within the main text. However, I recommend a complete revision by a native English speaker

- Throughout all figure captions be consistent in the description of your animal groups. Sometimes the age of mice is specified, sometimes not.

- The authors are generally describing a set of very interesting and valuable data for this field of research! However, the lack of a general overview and clear experimental strategy is partially making it hard to follow.

Specific comments:

l.26: “biosafety betacoronavirus” is no scientifically correct term. It is a BSL-2 betacoronavirus

l.28: what is meant with “pathological test”? Please find a specific term. This leads to missunderstandings

l.34: I do not see the aspect of “low cost” being of such importance that it needs to be stated in the abstract. Money shouldn’t be a justification for the value and applicability of an animal model

l.61: OE is susceptible to SARS-CoV-2 infection and not the other way around

l.83: It is the order of Nidovirales (delete order in Nidoviralesorder)

ll.90-101: the whole paragraph describes your results. This is an extensive description of results for an introduction. Please keep it shorter and only give an overview of key findings.

l.106: and throughout the whole manuscript: Don’t write 20µl of 10^4.66 TCID50/0.1ml but give a precise titer per animal! Since you know the volume and titer/volume and both are consistent for all animals, calculate the titer/animal.

l.107: Day of 100% mortality does vary between the two age groups. Should be mentioned here.

ll.111: No differences between sexes but between ages. This finding should be mentioned here or above.

ll.112: In the figure caption it is described that only 6 week old mice were used for this test. Also for the description of the animal experiments, is there a reason to do some experiments with both age groups, some with 6 week old, some with 3 week old mice? Rational should be explained somewhere.

l.132: Use another word than “propagated”. Which part of the brain?

ll.140: Figure S2 is mentioned here, but not that it also shows that you observed no changes in peripheral tissues of infected mice. This observation with referring to Figure S2 should be included accordingly.

l.147: Again, why now 3 week old mice?

l.151: Was that the nasal cavity also screened in this time course after infection (as described in table S1)? I think this table and the Figure 3A,B are highly interesting and can elucidate mechanisms of neuroinvasion. To have a complete picture, it would be perfect if the nasal turbinates would be included in the time course after infection.

l.152: Use another word than “suffering”

l.165: You observe reduced mortality but still the same neurological symptoms? Were functional assays (burying etc.) also performed with these groups?

l.166: You describe RNA data, which says nothing about infectious virus. Please re-phrase.

ll.222: This statement is not true. Syrian golden hamsters are currently the most widely used animal model for SARS-CoV-2 and also show neuroinvasion to a certain extent. Please also discuss the advantage of your model compared to the SARS-CoV-2 hamster model

ll.227: To me this hypothesis mostly seems to be based on extrapolation. How can a model with another pathogen be used to study efficacy of therapeutic interventions? No direct comparions were performed that can proof the applicabilty of the model in this respect. Consider toning down the use of your model in this respect.

ll.291: What is the specific use for this model in the future?

ll.308: How were the polyclonal antibodies prepared? Either refer to previous publications or describe more details here. Were animals immunized? Which animals, which antigen, including adjuvants etc.?

ll.318: Please be more precise about the use of animals and different sub-experiments. It is very confusing to get an overview which and how many animals were used to answer the different questions, addressed by different experiments. See also general comment

l.634: Please change “viral loads” to “viral genome loads”

l.635: Why is it n=4 for titrations and n=6 for PCR?

l.644: Sacrificed at what day post infection?

l.654: What is the added value of the PBS group compared to the untreated animals? Why do all groups have different group sizes?

l.656: Which of the above mentioned groups is subset (B) describing? And How can it be 6 mice per time point (3 in total) if one treatment group consists of maximum 16 animals?

ll.657: Which animals are described in (C) and (D)?

l.691: n=7 is described here, whereas in caption of Figure 1 it is n=5. Or are these separate experiments?

Figure 1 D-L: I am missing the mock-controls or are they reflected in the 0 dpi data? Don’t you assume a behavioral adaptation of animals over time and wouldn’t it be ideal to include measurements for mock-controls at all time points?

Figure 2 A,B: Use a Log10 y-axis for both graphs

Figure 3 A,B: Very impressive images. Was this also done for the 1 dpi time point and for the nasal cavity?

PLOS authors have the option to publish the peer review history of their article (what does this mean?). If published, this will include your full peer review and any attached files.

Reviewer #1: No

Reviewer #2: No

Reviewer #3: No
---

## [Decision Letter · Decision Letter 1]

2 Jun 2022

Dear Dr. He,

Thank you very much for submitting your manuscript "PHEV infection: a promising model of betacoronavirus-associated neurological and olfactory dysfunction" for consideration at PLOS Pathogens. As with all papers reviewed by the journal, your manuscript was reviewed by members of the editorial board and by an independent reviewer. The reviewers appreciated the attention to an important topic. Based on the reviews, we are likely to accept this manuscript for publication, providing that you modify the manuscript according to the editors recommendations below.

The authors have addressed the reviewers' comments  concerning the  experimental design and the presentation of the data accurately. However, the comments concerning the interpretation of the results in the context of SARS-CoV-2 infection are only partly addressed. Although the introduction and abstract have been adjusted, the authors summary and discussion are still largely written in the context of SARS-CoV-2. In a new version of the manuscript this should be adjusted.

1. In the authors summary the focus is the usefulness of PHEV infections in mice as a model to study the pathogenesis neurological complications of SARS-CoV-2. Please modify this according to the previous comments of the reviewers.

2. The discussion is too long and too much focused on SARS-CoV-2. In addition, it lacks a thorough discussion on the limitations of experimental PHEV infection in mice as a model for SARS-CoV-2 associated CNS complications. It is for example not included that there is a large difference in the ability of SARS-CoV-2 and PHEV to spread throughout the CNS. Although SARS-CoV-2 virus antigen/RNA is occasionally detected in the CNS of humans (or relevant animal models), it does NOT replicate/spread efficiently throughout the CNS causing a lethal encephalitis. This is an crucial difference, and for this reason the mechanism of SARS-CoV-2 associated neurological disease (including anosmia and ageusia) are likely very different.

[1] A letter containing a detailed list of your responses to all r comments, and a description of the changes you have made in the manuscript.

mportant additional instructions are given below your reviewer comments.

Sincerely,

Debby van Riel

Guest Editor

PLOS Pathogens

Ron Fouchier

Section Editor

PLOS Pathogens

Kasturi Haldar

Editor-in-Chief

PLOS Pathogens

orcid.org/0000-0001-5065-158X

Michael Malim

Editor-in-Chief

PLOS Pathogens

orcid.org/0000-0002-7699-2064

Data Requirements:

Reproducibility:

References:

---

## [Editor Report · Decision Letter 2]

10 Jun 2022

Dear Dr. He,

We are pleased to inform you that your manuscript 'PHEV infection: a promising model of betacoronavirus-associated neurological and olfactory dysfunction' has been provisionally accepted for publication in PLOS Pathogens.

Best regards,

Debby van Riel

Guest Editor

PLOS Pathogens

Ron Fouchier

Section Editor

PLOS Pathogens

Kasturi Haldar

Editor-in-Chief

PLOS Pathogens

orcid.org/0000-0001-5065-158X

Michael Malim

Editor-in-Chief

PLOS Pathogens

orcid.org/0000-0002-7699-2064

---

## [Editor Report · Acceptance letter]

22 Jun 2022

Dear Dr. He,

We are delighted to inform you that your manuscript, "PHEV infection: a promising model of betacoronavirus-associated neurological and olfactory dysfunction," has been formally accepted for publication in PLOS Pathogens.

Best regards,

Kasturi Haldar

Editor-in-Chief

PLOS Pathogens

orcid.org/0000-0001-5065-158X

Michael Malim

Editor-in-Chief

PLOS Pathogens

orcid.org/0000-0002-7699-2064